# Effect of Retarders on the Reactivity and Hardening Rate of Alkali-Activated Blast Furnace Slag Grouts

**DOI:** 10.3390/ma16175824

**Published:** 2023-08-25

**Authors:** Faten Souayfan, Emmanuel Rozière, Ahmed Loukili, Christophe Justino

**Affiliations:** 1École Centrale Nantes, Nantes Université, CNRS, GeM, UMR 6183, F-44000 Nantes, France; 2Soletanche-Bachy, Chemin des Processions, F-77130 Montereau Fault Yonne, France

**Keywords:** alkali-activated slag, set modifiers, organic acid, water content, viscoelastic properties, reaction advancement, mechanical properties

## Abstract

Sodium silicate-activated slags have the potential to harden quickly, which limits their practical use in grouting and deep soil mixing works. The open time of grouts is defined as the time period when their rheological properties allow their storage, pumping, and injection into the soil. In this work, the impact of the H_2_O/Na_2_O ratio and two acids (citric and boric acid) on the reactivity and hardening rates of slag-based grouts was studied. The H_2_O/Na_2_O ratio had a minimal impact on the open time but prolonged the setting time, as observed by ultrasonic characterization. Both acids were effective in delaying the structuration time, as revealed by oscillatory rheology and reaction advancement; however, they caused a decrease in the elastic modulus. Adding the acids resulted in a decrease in the pH of the medium, which may be linked to the extended open time. The analysis of the ion concentration of Ca, Si, and Al disclosed the mode of action of the two retarders.

## 1. Introduction

Considering current technical and environmental challenges, the use of alkali-activated materials in civil engineering appears to be a promising alternative to conventional hydraulic binders [1,2]. The use of blast furnace slags has gained attention in recent years due to the potential environmental benefits and improved performances [3]. The use of sodium silicate as an activator for slag-based materials is likely to provide high-performance materials in the hardened state; however, it can result in a quick setting process that poses a challenge for their practical use in the fresh state. This rapid setting can be attributed to several factors, including the chemical composition of slag [4], its particle size distribution [5], and the concentration of the activator [6]. However, the use of alternative activators such as NaOH or Na_2_CO_3_ can cause a delay in the setting process compared to sodium silicate [7]. Hung et al. [8] found that the initial and final setting times of alkali-activated slag were significantly reduced under lower SiO_2_ and Na_2_O dosages. Dai et al. [9] conducted a study on the impact of the fly ash partial replacement, activator, and water-to-solid ratio on the structuring of slag-based materials using oscillatory rheology. Their findings revealed that the addition of fly ash leads to a delay in the structuring process, while an increase in the water-to-solid ratio induced an acceleration at an early age. Various patents and studies have shown that admixtures can effectively delay the setting of activated slags [10,11,12,13]. A recent study showed the use of borax and citric acid was more effective than sucrose in extending the setting time of alkali-activated slags [14]. The research revealed that both sucrose and citric acid had an adverse impact on the compressive strength. However, it was reported that citric acid had an accelerating effect on alkali-activated fly ash material [15]. These retarders act differently to delay rapid setting. Citric acid lowered the initial pH of the medium and formed a set retarding barrier layer on the particle surfaces by precipitating calcium carboxylate [16]. The addition of boric acid slowed down the dissolution rate of silicates and aluminates [17]. Sodium tetraborate can extend the setting time by altering the gel structure and mortar properties through the transformation of Al-O-Si bonds to B-O-Al-O-Si [18,19]. Another study stated that the incorporation of borax produced gels, covered the surface of the slag, and slowed down the hydration reaction rate [14]. While the design of cement-based materials requires various retarders to achieve optimal results, these same retarders cannot be used in the same conditions for alkali-activated materials due to certain limitations [14]. This is primarily due to the diverse composition of alkali-activated materials and initial mixture conditions. Some retarders can delay the reaction progress of particular mixtures while having no impact on others, highlighting the need for a good understanding of the initial mix to select the most suitable retarder. This makes it difficult to find a retarder that can be stable in highly alkaline conditions while also effectively delaying the reaction. Previous studies have focused on the changes in the initial and final setting times, and they have not paid enough attention to the open time and diluted systems, such as grouts. The focus of this study is to examine how retarders affect the open time of grouts, as well as their impact on the mechanical properties in the hardened state. Grouts are relatively dilute mixtures, and they are expected to meet several specifications in the fresh and hardened states, hence the need for a comprehensive experimental study. Fresh grouts should be stable during storage and before injection, fluid enough to obtain a uniform and homogeneous soil–grout mixture, and these properties should be maintained during the whole duration of the soil mixing process [20,21,22]. There are two distinct periods to consider. The first one is the open time, which begins when the grout is prepared. During this time, the grout must maintain a specific viscosity that is suitable for pumping and injection. It is noteworthy that this practical working time is different from the setting time, which refers to the time required for the grout to shift from a fluid state into a solid state with a minimum compressive strength value. In the hardened state, grout is designed to reach specified mechanical and durability properties. This article presents a comprehensive study on the impact of the H_2_O/Na_2_O ratio and two acids (citric and boric acid) on the structuring kinetics of slag-based grout systems. The effect of the water content is studied, as it is a key parameter in controlling the properties of alkali-activated materials in the fresh state [23,24]. The investigation includes the whole material preparation process from the fresh state of the material to its hardened state. The behaviors of fresh and hardened materials were investigated through isothermal calorimetry and dynamic rheology in the linear viscoelastic range. The role of two retarders on the pore solution composition was investigated, and the results of chemical analyses were correlated with the evolution of rheological properties. The effect of retarders on mechanical properties was also studied.

## 2. Materials and Methods

### 2.1. Materials

The blast furnace slag used in this study was provided by Fos, France. The chemical composition of the ground granulated blast furnace slag (GGBFS) is described in Table 1. The median particle size of the GGBFS was 10 µm, its density was 2.90, its activity index was 1 at 28 days, and its specific surface area was 0.45 m^2^/g, as determined by N_2_ sorption through the Brunauer–Emmett–Teller (BET) method. Its particle size distribution is presented in Figure 1.

To prepare the activation solution, the sodium silicate solution was diluted using demineralized water while maintaining a fixed SiO_2_/Na_2_O ratio of 1.7 and a dry content of 44%. The molar ratio of SiO_2_ to Na_2_O in the activator was in line with literature recommendations to ensure effective precursor dissolution [25]. The sodium silicate solution was specially developed to contain a high Na_2_O content, which would minimize the need for additional NaOH during the mixture preparation process. It is noteworthy that the mix proportions (Table 2) were carefully selected to optimize rheological and mechanical properties based on previous studies [23,26,27].

The study of retarders is divided into two axes: the effect of organic acids and water content by varying the H_2_O/Na_2_O ratio. Four mixtures listed in Table 1 were designed to determine the impact of H_2_O/Na_2_O levels on the setting and hardening process. The H_2_O/Na_2_O molar ratio has been calculated by considering the total number of moles of H_2_O, which includes demineralized water and the water fraction in the sodium silicate solution. The number of moles of Na_2_O includes the sodium in NaOH and the amount of sodium in the solid fraction of the sodium silicate solution.

The sodium silicate solution-to-precursor ratio was kept constant while the water content was modified. The water-to-solid ratio (%wt) was calculated considering the total water content in the mixture, including the added demineralized water, the water from the activation solution, and the total amount of solid, including the precursor, solid fraction of the sodium silicate solution, sodium hydroxide, and bentonite. The percentage of solid phases, in terms of volume, includes the volume of solid components such as NaOH, bentonite, slag, and the solid fraction of the sodium silicate solution. Bentonite was added to stabilize the material in the fresh state, based on a previous study [27]. The effect of two types of retarders (citric acid and boric acid) on the reaction progress of mixture S38 defined with an H_2_O/Na_2_O ratio of 38 was studied. The admixture contents were established from patent recommendations and previous studies [10,11,12,13]. The retarder was added after the preparation of the activation solution and just before the incorporation of the slag. Citric acid was used at 2%, 3%, and 4%, and boric acid at 2% and 3%, relative to the amount of slag in the grouts. The activation solution was prepared before adding the powders, which were then stirred in a high-shear mixer for 5 min. The initial pH of the medium was measured using a pH electrode. Due to the occurrence of an “alkali error”, which leads to an overestimation of the pH value, caution is necessary when considering the reliability of pH measurements in high alkali ion content [28].

### 2.2. Testing Procedures

#### 2.2.1. Isothermal Calorimetry

The short-term reactivity was assessed at a temperature of 20 °C using an isothermal calorimeter (TAM Air, TA Instruments, New Castle, DE, USA). Heat flow was measured, and the cumulative heat was obtained by integrating the measured flux. The measurements were carried out for the first seven days of reaction. The cumulative heat at time t (Q(t)) was extrapolated to determine the ultimate heat (Q∞). This was calculated by plotting the cumulative heat as a function of 1/√t and determining the intersection of this linear function with the *y*-axis [29].

#### 2.2.2. Scissometer

The scissometer is commonly used in soil testing to assess the cohesion of material under shear. It is defined by the French standard, NF P 94-112 [30]. The scissometer has been developed to measure the cohesion of fine soils such as clay, silt, and mud [31,32]. It allows for estimating the shear strength of soils under zero normal stress in an undrained situation. Its geometry and principle are close to rotating vane rheometry, which is widely used to characterize inorganic colloidal dispersions. The main advantage of this measurement technique is to limit wall slip effects. The torque (T) measured by the scissometer is directly proportional to the shear strength of the material (τ) through the coefficient, k, which depends on the blade geometry. The scissometer was equipped with interchangeable blades of different sizes, and the critical value could be read directly with a graduated ring set to zero before each measurement. Depending on the size of the vane and the maximum value of the graduated ring used, the scissometer allows the measurement of shear strengths of different orders of magnitude. The open time of grouts is defined as the time when the shear strength of the grout reaches 200 Pa.

#### 2.2.3. Rheometer

A Discovery Hybrid Rheometer from TA Instruments with vane geometry was used to perform dynamic rheological measurements at a constant temperature of 20 °C. The vane geometry was chosen to minimize disturbance of the material and prevent wall slip [33]. The cup and bob diameters were 30.36 and 26.198 mm, respectively, hence a shear gap of 4 mm. The mixtures were characterized in the fresh state using dynamic rheology in the linear viscoelastic range. The linear viscoelastic domain was established to ensure that measurements were non-destructive. Fresh materials were prepared and introduced into the bowl of the rheometer, and residual stresses were allowed to relax before each test. A pre-shear was applied at 200 s^−1^ for 60 s, followed by oscillation with a strain of 5% at a pulsation rate of 5 rad/s. After preconditioning, experiments were carried out at a strain of 0.01% and a pulsation rate of 10 rad/s [34,35]. The elastic modulus (G′), viscous modulus (G″), and loss tangent (tan δ = G″/G′) were analyzed within the first few hours. The short-term reactivity was assessed at a temperature of 20 °C using an isothermal calorimeter (TAM Air). Heat flow was measured, and the cumulative heat was obtained by integrating the measured flux. The measurements were carried out for the first seven days of reaction. The cumulative heat at time t (Q(t)) was extrapolated to determine the ultimate heat (Q∞). This was calculated by plotting the cumulative heat as a function of 1/√t and determining the intersection of this linear function with the *y*-axis [29].

#### 2.2.4. Young’s Modulus of Fresh and Hardening Materials

The Freshcon system was developed at the University of Stuttgart to analyze the setting and the hardening of cementitious materials from the velocity of compression and shear ultrasonic waves (P and S). This system enables the monitoring of Young’s modulus and the shear modulus of hardening material from an early age. This part of the study was conducted on mortar composed of 47% (by volume) alluvial siliceous sand (quartzite) of 0/2 mm and 53% grout.

#### 2.2.5. Elastic Modulus in Hardened State

The static modulus of the elasticity (E) of the mortar was evaluated on prisms (40 × 40 × 160 mm^3^) following NF EN 196-1 [36] after 7, 28, and 90 days. This test was performed on two samples of each mixture to investigate the effect of the retarders on the mechanical characteristics of the grouts.

#### 2.2.6. Compressive Strength

For the compressive strength determination, slag cement grouts were cast in cylindrical molds with a diameter of 70 mm and a height of 140 mm, in accordance with the NF EN 12390-1 standard [37,38]. Samples were kept in their molds in a humid room at 20 °C until the time of testing due to the high water-to-solid ratio, which hindered the demolding of several samples at an early age. This procedure was adopted to ensure uniform conditions for curing and demolding. The compressive strength testing was carried out on a 100 kN press at 28 days. The load rate was 1.9 kN/s until failure [39]. The strength data allowed for investigating the effect of the H_2_O/Na_2_O ratio on the mechanical characteristics of the grouts.

#### 2.2.7. Pore Solution Composition

The time evolution of Al, Si, and Ca ionic species concentrations in the pore solution was monitored through inductively coupled plasma emission spectroscopy (ICP/ISP). Pore solutions of the grouts were centrifuged at the age of 5, 60, 120, and 150 min. The initial ionic concentrations of the activator solution were determined by ICP analysis. The obtained pore solution was filtered by a syringe connected to a disposable 0.45 µm syringe filter. The pore solution was diluted by the ratio of 1:1000 with pure water to determine the concentration elements in the pore solution.

## 3. Results and Discussion

### 3.1. Influence of H_2_O/Na_2_O Ratio

#### 3.1.1. Reaction Advancement and Structuration

In Figure 2, the evolution of heat release in slag-based grouts is presented at various H_2_O/Na_2_O ratios. The heat flow was assessed using isothermal calorimetry and standardized based on the mass of the grout (the total mass of the mixture). Additionally, we attempted to normalize it with respect to the mass of the slag to investigate the impact of dilution on slag reactivity. The resulting curves showed no significant influence of water-to-solid on the cumulated heat release. Therefore, the data are presented relative to the overall mass of the grout. The impact of the ratio on the heat flux is significant. The intensity of the initial peak decreases with the water content, and the ‘dormant period’ extends as the medium is diluted. This stage occurs between particle dissolution and gel growth. Subsequently, the main peak is observed, corresponding to the maximum of the silicate reaction. Increasing the H_2_O/Na_2_O ratio delayed this peak, indicating that this ratio had a direct effect on the progress and formation of hydration reaction products in these grouts.

The rheological analysis indicates that a higher H_2_O/Na_2_O ratio results in a higher initial modulus G′, as shown in Figure 3. Furthermore, the addition of water does not delay the early development of stiffness. The increase in the H_2_O/Na_2_O ratio could enhance the dissolution of the initial solid phase due to higher concentration gradients. This, in turn, results in reaching a critical ion concentration responsible for the initial increase in G′, which is observed at a low H_2_O/Na_2_O due to a higher activator concentration. It is noteworthy that the initial variation of G′ occurs during the slag dissolution phase prior to the induction phase. This evolution of G′ is related to the wetting of particles and dissolution of slag and partly due to the formation of initially dissolved silicate units and their interactions with Ca and Na [9,40]. To prevent the hardening of the studied materials in the rheometer, G′ could not be monitored in the later stages.

In order to establish a correlation between the mechanical properties at an early age and the reaction chemistry, Young’s modulus E was determined using the FreshCon method, as illustrated in Figure 4. In all the cases, the setting process is initiated rapidly after the constituents are mixed, followed by a pseudo-plateau in Young’s modulus and then a second acceleration that coincides with the onset of the second calorimetry peak (Figure 5). The evolution of Young’s modulus was more pronounced in S38. This mixture led to a higher modulus compared to other materials, which also exhibited some variations in modulus values and setting times. Notably, an increase in the H_2_O/Na_2_O ratio resulted in longer setting times and a prolonged constant plateau phase. By integrating heat flow and modulus measurements, a correlation between the physicochemical mechanisms is revealed, as presented in Figure 5.

The initial increase in Young’s modulus can be attributed to the early formation of C-A-S-H gel [41,42], as indicated by the simultaneous first peak in the heat flux. At this stage, G′ actually reached high values, and it was necessary to remove the sample from the rheometer. The second peak in the heat flow was accompanied by a more significant increase in Young’s modulus.

#### 3.1.2. Mechanical Properties in Hardened State

Feret’s equation was used to distinguish the impact of the binder type and the initial porosity of the mixture (Equation (1)):(1)$$fc=k*{\left(\frac{C}{C+W+A}\right)}^{2}$$
where k depends on the properties of the aggregates and cement, which are assumed constant, whatever the proportions. C, W, and A are the volumes of the cement, water, and air, respectively. Feret’s equation can be adapted to analyze the behavior of the studied alkali-activated materials, considering:
C as the initial volume of reactive solids in the material (the precursor and solid fraction of the activation solution);W as the total water content in the mix;A, as the air volume, is considered negligible in the material (confirmed by measurements of the initial density).

The experimental compressive strength and the values calculated by Feret’s equation are plotted in Figure 6. The least squares method was used to determine the line of best fit for the experimental data. The evolution of the strength calculated by Feret’s equation is globally consistent with experimental values, indicating that the strength of the slag-based grouts was proportional to the initial porosity, as observed in cement-based materials. Thus, the influence of the H_2_O/Na_2_O ratio and alkaline conditions on the reactivity of slag activated by sodium silicate was minimal within the studied H_2_O/Na_2_O range (H_2_O/Na_2_O = 38–62). Feret’s approach shows that these grouts behaved like Portland cement-based materials with respect to initial porosity and binder properties, regardless of the water content. However, these grouts showed a rapid initial setting even when adjusting the H_2_O/Na_2_O ratio. This ratio delayed the advancement of the reaction over time, but it did not provide any practical advantages in terms of open time.

### 3.2. Influence of Retarders on Setting Kinetics

#### 3.2.1. Effect on the Flow of Heat Released

Two sets of retarders were tested for the L38 grout (Ref): boric acid at 2% (B2%) and 3% (B3%) and citric acid at 2% (C2%), 3% (C3%), and 4% (C4%), relative to the mass of slag in the mix. The reaction of the slag and activation solution undergoes several stages, including dissolution, induction, acceleration, and deceleration [2]. The impact of additives on these stages can be observed in Figure 7. The addition of boric acid delayed the peaks observed for the reference grout and decreased their respective intensities while increasing their width (Figure 7a). The addition of citric acid delayed the main silicate peak. When using 3% and 4% citric acid and 3% boric acid, two consecutive peaks were observed, followed by an extended induction phase before the main silicate peak appeared. This demonstrates that the retarders influence the reaction advancement differently depending on their contents and compositions. The addition of 4% citric acid caused a higher delay in the reaction compared to other combinations. Despite a peak after 10 days, the heat emitted from the C4% mixture remained low, even after seven days. Citric acid did not stop the reaction completely but significantly slowed its advancement when compared to other combinations. The released heat indicates that acids have a significant impact on the advancement of the alkali-activated reactivity of slag.

#### 3.2.2. Structural Build-Up Properties

In Figure 8, the scissometer is used to monitor the stiffness of different mixtures. The use of acids boosted the initial stiffness rate when compared to reference mixture S38. However, increasing the percentage of additives in the blend eventually slowed down the development of the stiffness. If the stiffness threshold of 200 Pa is taken into consideration, the open time can be extended from 80 min to 185 min by adding 3% of boric acid (B3%) compared to the reference mixture. The gel time determined from the scissometer is represented in Table 3. Nonetheless, relying solely on the scissometer stiffness rate may not be the most appropriate way to assess the material structure since it only provides discontinuous characterization. These data do not allow us to understand the phenomena occurring when the material shifts from the fresh state to the hardening stage. Additionally, the test is destructive, requiring the fabrication of a significant amount of grout. For each measurement time, a separate sample is needed. Therefore, it is useful to study the impact of both additives on material structuring by employing oscillatory rheology.

The initial focus of this part consists of analyzing how the use of admixtures affected the development of the elastic modulus. Subsequently, the analysis deals with the evolution of the concentrations of aluminum ions (Al), silicon (Si), sodium (Na), magnesium (Mg), and calcium (Ca) to monitor the advancement of the reaction and correlate observed evolutions. Figure 9 illustrates the elastic modulus obtained from oscillatory rheology measurements. The arrows represent the gel time* of the grouts. Gel time* represents the gel time determined from the evolution of G′ in Figure 9b. Two tangent lines were drawn on each G′ curve with a logarithmic time scale. The time when the first tangent line (which corresponds to the development of the first modulus of rigidity) intersects with the second tangent line (which corresponds to the multiplication of the percolation paths) was defined as the gel time* of the grouts. The presence of retarders modified the material structuring and initial stiffness, as evidenced by the higher initial modulus values (Figure 9b). These results are consistent with those obtained using the scissometer (Figure 8). Modulus G′ exhibited a faster increase with 4% citric acid, but this material had a slower growth rate than the mixture with 3% boric acid, indicating that the nature of the retarder and its content affected the structuring kinetics differently.

Figure 10 depicts the impact of structuring kinetics on the time-dependent evolution of tan δ. The loss tangent, tan δ, is defined as the ratio of the viscous modulus G″ to the elastic modulus G′ (tan δ = G″/G′). First, tan δ decreased because G′ varied more quickly than G″, and it was always lower than 1 (G′ > G″). A decrease in the loss factor can be attributed to the gradual gelation of the material [43], and then a peak in tan δ appeared. Several authors have mentioned that this peak presents a criterion of vitreous phase transition, which can be associated macroscopically with the gel time [9,24,35]. The value and time of this peak were influenced by the type and proportion of the retarder used, and the time of the tan δ peak increased with the acid content. The grout containing 4% citric acid manifests a longer time of the tan δ peak than the other mixtures, signifying that it maintained its viscoelastic state for a prolonged period and thus delayed the formation of a more solid network. The time corresponding to the peak in tan δ is compared to the gel time deduced from the scissometer measurements and oscillatory rheology in Table 3. The G′ curves were also used to define an indicator of the gel time. A rheometer would be a more convenient way to characterize the grout’s properties, as measuring the time with a scissometer required tracking multiple spots and using larger quantities of grout.

According to the results presented in Table 3, the evolution of the gel time and maximum tan δ time follows the same trend, except for the 3% boric acid grout, which has a longer gel time than the 4% citric acid grout but a shorter maximum tan δ time. The pH of the medium had no linear effect on the gel time and the maximum tan δ time, indicating that the mode of action of retarders depends on their composition. Additionally, there was a significant difference between the gel time and maximum tan δ time. The gel time values obtained from oscillatory rheology and the scissometer were found to be similar, validating the method of determining the gel time from the intersection of two tangents to the G′ curve. Thus, two characteristic times can be defined based on the viscoelastic properties’ evolution to characterize the rheological behavior of the grout: a gel time (gel time*), during which the grout maintains a suitable viscosity for processing, and a time corresponding to the phase change (max. tan δ).

To understand better the reaction mechanisms, it was necessary to monitor the concentration of ionic species (Al, Si, and Ca) in the pore solution over time through inductively coupled plasma emission spectroscopy (ICP/ISP). The Al, Si, and Ca concentrations and G′ evolution are presented in Figure 11. The first point of the curves corresponds to the concentration of elements in the activation solution. The evolution of these concentrations reflected the amounts of ions remaining in the interstitial solution after the consumption of the elements were released from the precursor to form the reaction products. The Si concentration showed higher values than Ca and Al since sodium silicate was used as the activation solution. Regarding the development of the initial value of elastic modulus G′, it is worth noting that the rise in G′, resulting from the addition of the acids, can be attributed, at least in part, to the consumption of the initial Si. This is supported by the observation of a decrease in the Si content after 5 min of mixing in these mixtures.

The rate of change in ion concentration varies among mixtures depending on their dissolution and consumption rates. In the case of the reference mixture without acid, the Si concentration underwent a small fluctuation over time, indicating that its dissolution and consumption rates were approximately equivalent. When acid was present in the mixtures, the concentration of Si measured 5 min after mixing was lower than the initial value, indicating that Si was consumed right after mixing. The concentration of Ca and Al ions also followed a similar pattern. Since they were absent in the activation solution, their initial concentration was zero, but it rapidly increased after mixing and continued to increase at varying rates, similar to modulus G′. The reference mixture showed a faster increase in concentration compared to the other two mixtures, suggesting a delay in the dissolution (or bonding) of Al and Ca from the precursor compared to the reference mixture. The dissolution of Al appears to be blocked until 1 h after mixing; after that, the concentration increased slowly. Overall, the small evolution of modulus G′ over time indicated that the consumption of ions for the formation of the reaction products was minor at this point in the reaction. It should be noted that these concentration measurements were possible within the range of open time. For the 3% boric acid mixture, one measurement was performed at a time (220 min), exceeding the open time or what is called ‘gel time’. The concentration of three elements decreased, more strongly for Al and Ca than Si. Ca and Al can be combined with Si to form hydrated calcium aluminosilicate hydrates. At this point, the concavity of the G′ curve started to change. It can thus be deduced that as long as dissolution outweighs ion consumption, the open time can thus be maintained. The use of both acids resulted in a reduction of the dissolution rate, leading to a delay in the formation of the reaction products. This decrease in the dissolution rate may be attributed to a decrease in the pH of the initial medium, as indicated in Table 3. Furthermore, these acids exhibited distinct mechanisms in delaying the setting process. Research has shown that citric acid delays the reaction by decreasing the initial pH of the medium and forming a layer of calcium carboxylate precipitation on the particle surfaces, which acts as a barrier against setting [16]. Figure 11 shows that the concentration of Ca in the 4% citric acid mixture underwent a slight decrease after one hour, indicating that the rates of calcium dissolution and consumption were approximately equivalent. Conversely, the other two mixtures exhibited a much greater increase in Ca ion levels, suggesting consumption of Ca ions and corroborating the citric acid retardation mechanism mentioned in the literature [16]. The retarding effect of boric acid could be due to the lower pH of the medium. This leads to a slower dissolution rate of silicates and aluminates over time, as observed in the findings discussed earlier [17]. However, the modification of the Al-O-Si bonds to B-O-Al-O-Si that changes the gel structure of ionic species, as mentioned in previous studies [18,19], could not be demonstrated from the evolution of the concentrations. The level of boron concentration in the mixture remains nearly constant during monitoring time, as demonstrated by the monitoring data presented in the Appendix A (Figure A1).

Based on the experimental results, the impact of retarders on the structuration process can be understood by examining their effects on various stages of the reaction. Initially, the presence of retarders altered the dissolution of several ions (Al, Si, and Ca), thereby influencing the evolution of G′. The reduction in the dissolution rate resulted in the retardation of the coagulation mechanism and led to a prolonged gel time. The extension in the increase of tan δ is directly associated with the delay in reaching the critical concentration, which, in turn, delays the condensation mechanism. This correlation is supported by the significant decrease in the ionic concentrations of Al, Ca, and Si, coinciding with the initiation of the increase in tan delta (as observed in the case of the mixture containing 3% of boric acid in Figure 11).

#### 3.2.3. Influence of Retarders on Mechanical Properties

The elastic modulus E at 7, 28, and 90 days is presented as a function of the cumulated heat deduced from isothermal calorimetry in Figure 12. A linear correlation can be observed for most of the studied materials. The addition of retarders led to a decrease in the elastic modulus (Young’s modulus, E). While the heat release after 28 and 90 days was slightly affected, the modulus significantly decreased. Specifically, the material with 4% citric acid had not gained significant stiffness at 7 days, but it reached a Young’s modulus of 7 GPa after 28 days, indicating that the reaction significantly advanced within the first month. The highest heat release was observed after 28 and 90 days for this mixture, but it also had the lowest modulus. This suggests that used retarders did not completely inhibit the reactivity of the slag, but they altered the resulting products and microstructure, and thus the mechanical properties. For example, calcium carboxylate formation was previously observed with citric acid [16].

## 4. Conclusions

The aim of this study was to investigate the impact of retarders on slag-based grouts. The study focused on the effects of the H_2_O/Na_2_O ratio and retarders, such as citric acid and boric acid, to slow down the structuring of activated slag.
The H_2_O/Na_2_O ratio had a limited effect on the initial stiffness development of the studied mixtures during the first two hours and, therefore, on the open time. An increase in the H_2_O/Na_2_O ratio leads to enhanced dissolution of the initial solid phase due to higher concentration gradients. This process may result in reaching a critical ion concentration, which is responsible for the initial increase in G′, which occurred simultaneously in the mixtures with lower H_2_O/Na_2_O ratios;However, it prolonged the setting time, as seen through ultrasonic characterization. The evolution of the compressive strength showed that these grouts behaved like cement-based materials, with respect to the initial porosity and binder properties, regardless of the water content;The rheological behavior of the grout, characterized by oscillatory rheology, can be characterized by two distinct times: the gel time when the grout maintains a suitable viscosity for processing and a time corresponding to the phase change (max. tan δ);Both acids were effective in delaying the structuration time and modifying the overall reaction progress, as revealed by oscillatory rheology and scissometer results. Increasing the percentage of boric acid from 2% to 3% and citric acid from 2% to 4% extended the open time by 80 min. However, the introduction of 3% and 4% boric acid and citric acid led to a decrease in Young’s modulus from 15 GPa to 12 GPa and 10 GPa, respectively, as compared to the reference;The analysis of Ca, Si, and Al ionic concentrations disclosed the mode of action of two retarders. Citric acid was found to delay the reaction by lowering the initial pH of the medium and potentially forming a layer of calcium carboxylate precipitation on the particle surfaces. The retarding effect of boric acid could be due to the lower pH of the medium;Finally, in order to find a balance between the open time and compressive strength, it is necessary to optimize the retarder contents in the mixture.

## Figures and Tables

**Figure 1 materials-16-05824-f001:**
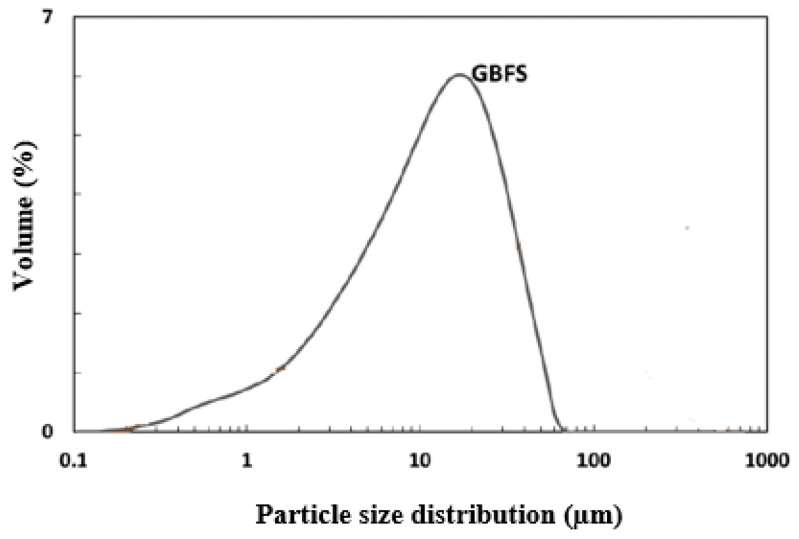
Particle size distribution of slag.

**Figure 2 materials-16-05824-f002:**
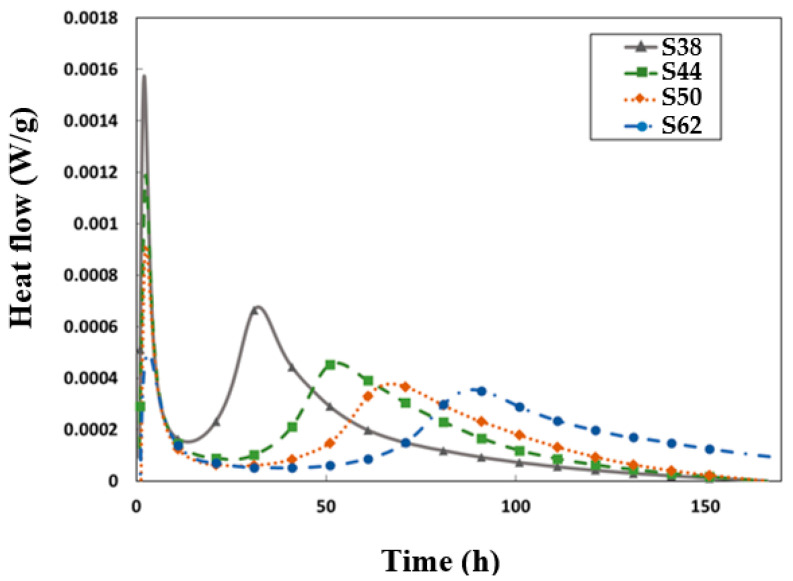
Effect of H_2_O/Na_2_O on the heat flow.

**Figure 3 materials-16-05824-f003:**
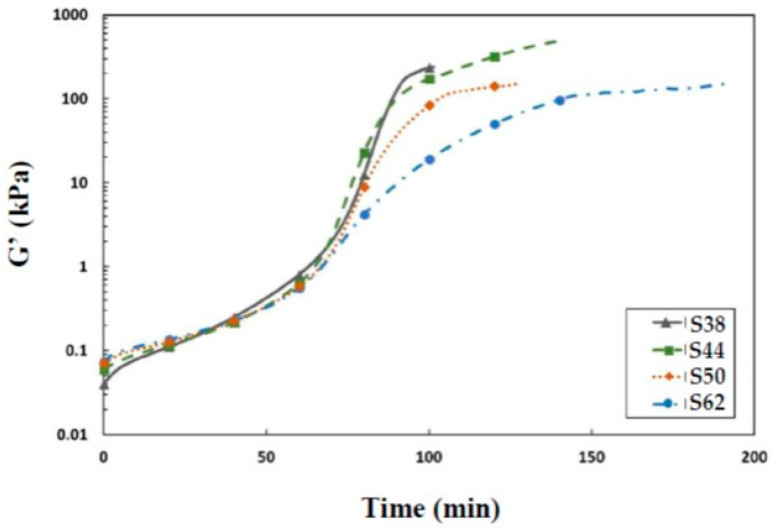
Evolution of G′ as a function of time.

**Figure 4 materials-16-05824-f004:**
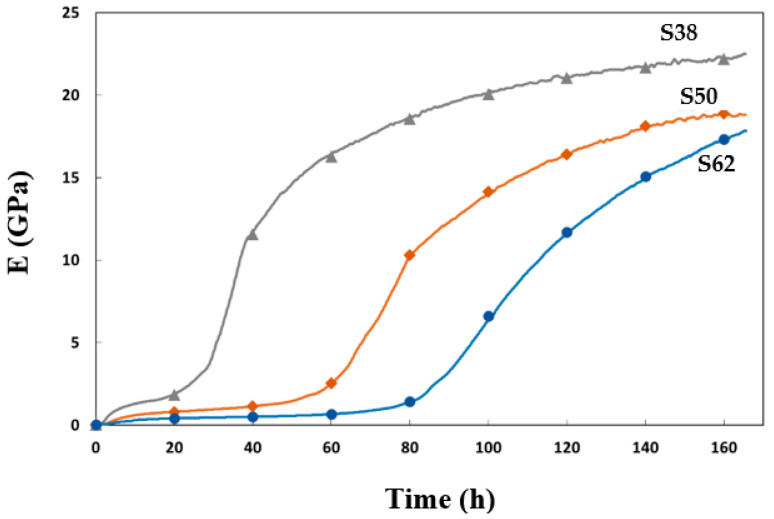
Evolution of Young’s modulus E of slag-based grouts.

**Figure 5 materials-16-05824-f005:**
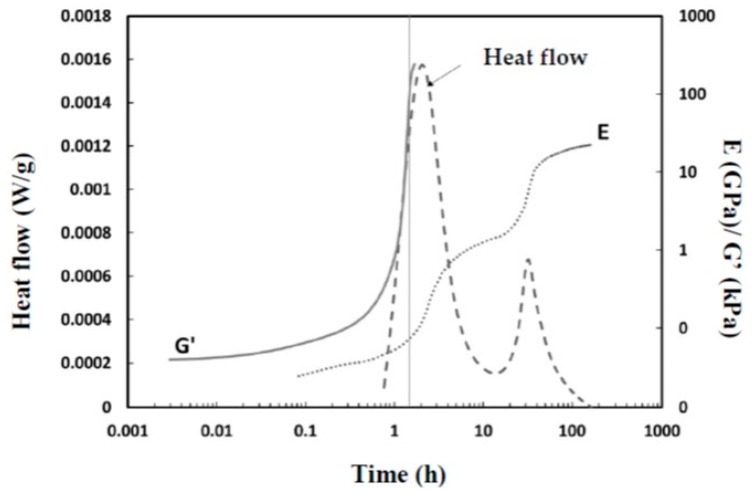
Time evolution of heat flow and Young’s modulus E and G′ of the S38 mixture.

**Figure 6 materials-16-05824-f006:**
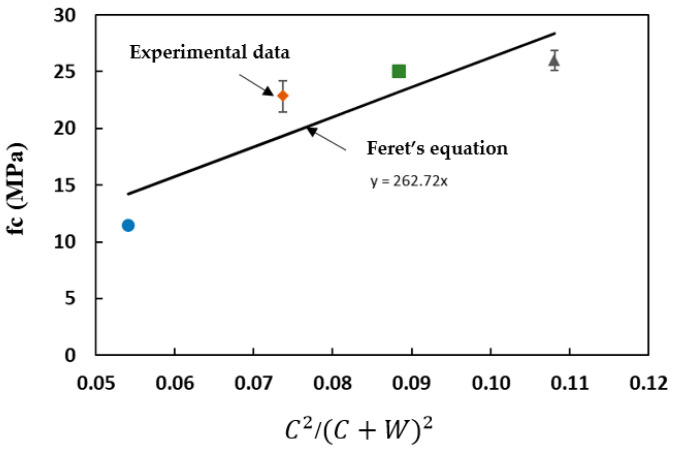
Comparison of experimental compressive strengths and values calculated by Feret’s equation at 28 days. Blue circle: S62. Red diamond: S50. Green square: S44Green triangle: S38.

**Figure 7 materials-16-05824-f007:**
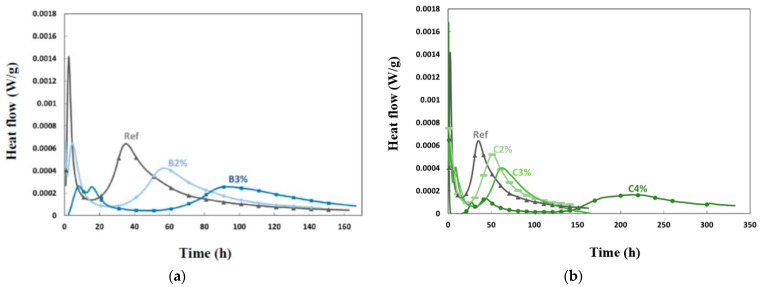
Effect of retarders on heat flow per mass of material. (**a**) Boric acid (**b**) Citric acid.

**Figure 8 materials-16-05824-f008:**
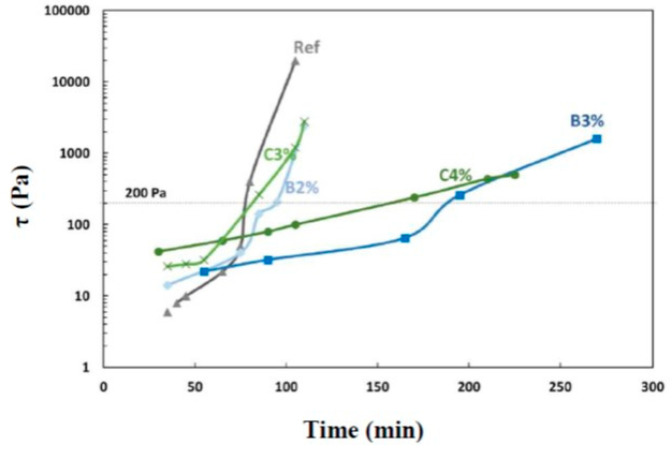
Evolution of the stiffness (Pa) measured using the scissometer.

**Figure 9 materials-16-05824-f009:**
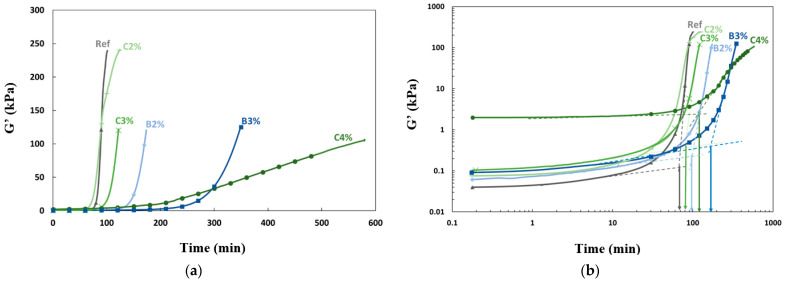
(**a**) Evolution of modulus G′ as a function of time (**b**) Determination of gel time.

**Figure 10 materials-16-05824-f010:**
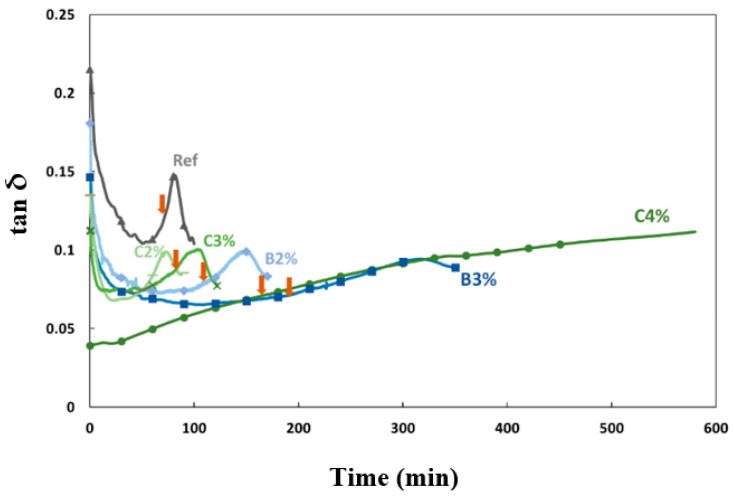
Tan δ as a function of time. The arrows indicate gel times.

**Figure 11 materials-16-05824-f011:**
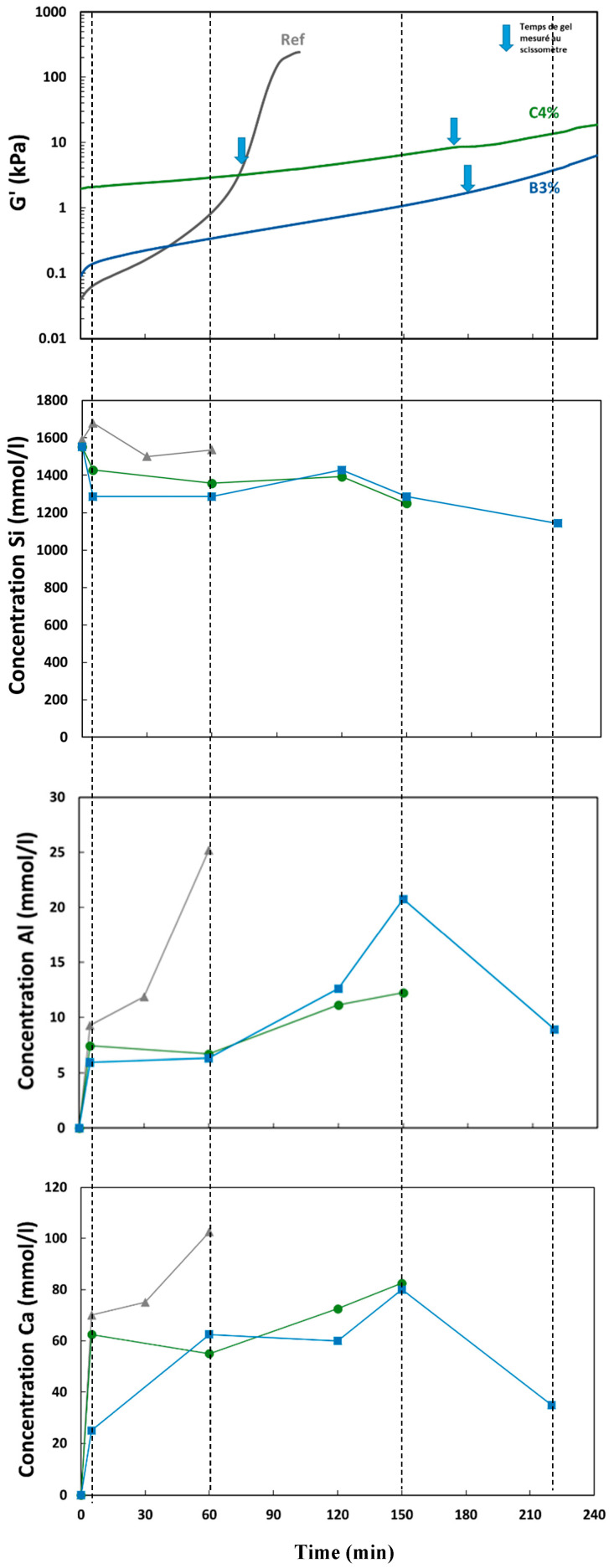
Evolution of G′ and concentration of Si, Al, and Ca.

**Figure 12 materials-16-05824-f012:**
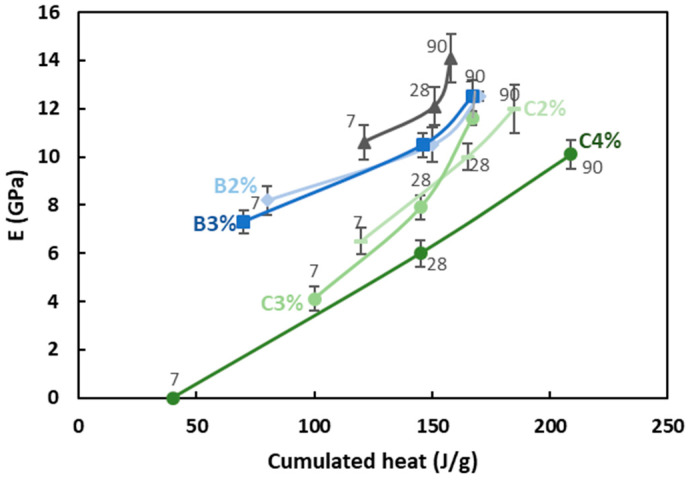
Evolution of elastic modulus E as a function of cumulated heat after 7, 28, and 90 days of curing.

**Table 1 materials-16-05824-t001:** Chemical composition (mass fraction %) of slag.

**(wt.%)**	**SiO_2_**	**Al_2_O_3_**	**CaO**	**Fe_2_O_3_**	**MgO**	**SO_3_**	**Cl^−^**	**TiO_2_**	**Na_2_O**	**LOI**
	37.2	10.5	43.2	0.6	7	0.1	0.01	0.5	0.6	ε

**Table 2 materials-16-05824-t002:** Compositions of the studied mixtures.

Mixtures		S38	S44	S50	S62
Materials	Density	Compositions (g/L)			
Slag	2.90	861	780	711	608
Na-silicate	1.55	348	315	288	246
NaOH	1.46	3.5	3.2	2.9	2.5
Bentonite	2.50	14	16	17	19
Water	1.00	476	526	567	623
Water/solid ratio (W/S %wt)		0.65	0.75	0.85	1.04
Volume fractions (%)					
Water		66	69	72	76
Solid phases		34	31	28	24
Molar ratio of H_2_O/Na_2_O		38	44	50	62
pH		13.25	13.22	13.19	13.12

**Table 3 materials-16-05824-t003:** Structuration time and pH of the medium for the different mixtures.

Formulations	Scissometer	G′ Evolution—Oscillatory Rheology		pH (±0.01)
	Gel Time (min)	Gel Time* (min)	Max. tan δ (min)	
Ref	75	70	82	13.25
B2%	95	100	150	13.02
B3%	185	180	315	12.76
C2%	Not measured	50	75	12.90
C3%	80	80	100	12.48
C4%	165	130	577	12.50

## Data Availability

Not applicable.

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
