# Peer review of "Effect of Retarders on the Reactivity and Hardening Rate of Alkali-Activated Blast Furnace Slag Grouts"

_materials, 2023, doi:10.3390/ma16175824_

Round 1

Reviewer 1 Report

·       Please provide the particle size distribution of slag as it is also very important for the properties of the BFS-based alkali-activated binders;

·       In Table 2, what does ‘solid’ and ‘solid phases’ stand for respectively? You should make the mixture proportion clearer and more understandable;

·       2.1.1., cumulative heat at time was extrapolated to determine, not the determine;

·       I am curious about if there are any more standards and papers related to scissometer instead of the single French standard NF P94-112;

·       In 2.1.6, please specify the curing condition while samples were in the moulds. Actually, samples should be demoulded after one day and cured under standard condition (normal temperature but high humidity chamber) or in water directly;

·       What’s the detail of the dilution of the pore solution as I think the dilution degree would also affect the measurements of the concentration of elements;

·        In Fig. 1., you should point out the unit of y-axis (heat flow with a unit at W/g), how did you get the values and W per unit mass (g) of what? These values are very important since different mixes had various amounts of solid phases and/or slag;

·       For the rheological analysis, you have to propose possible explanations for why different H2O/Na2O ratios led to similar stiffness in the early stage;

·       Line 218-222, the initial increase in the Young’s modulus should be associated with the dissolution of slag in the alkaline activator because this increase concurs well with the very first peak in the heat flux. But here you attributed to the formation of C-A-S-H. I think there could be some C-S-H gels forming at this time, but not C-A-S-H since they are supposed to form after 30-40 hours, corresponding to the second peak in the Fig. 1.

·       In the conclusion part, more of the related mechanisms behind the experimental results should also be presented.

·       Did you use duplicates or triplicates to conduct the isothermal calorimetry test?

·       ‘The addition of citric acid caused the first dissolution peak to appear earlier than in S38’, how did you find this? I found it’s not the case.

·       ‘Moreover the test is 283 destructive, which implies testing as much samples as measurement times thus repeatability issues.’ This sentence should be rewritten as it is confusing;

·       How many samples were involved in the 3.2.2.? I think it’s better to have values in Table 3 with standard deviations, otherwise it would not be so convincing;

·       Line 360-361: Please explain or give some possible reasons why the dissolution of Al appeared to be blocked;

·       In Fig. 10, you should label out the 7, 28, and 90 days in the figure.

It's better to invite a native speaker or professional native researcher to polish the English. 

Reviewer 2 Report

The authors have presented a topic of significance in a very systematic manner. Some minor modifications are suggested:

1. Kindly provide the LOI, activity index, and D50 of the slag used 
2. Kindly provide the basis for the selection of mix proportions - the studies by Rangan, Wallah, Bakharev, or Ramagiri can be used as reference
3. Kindly provide microanalytical results to show the change in the morphologies due to the reactions - there is an elaborate discussion on the same in the studies by Ramagiri et al. for the authors' reference
4. Kindly provide the stepwise effect of adding the admixtures on the destruction-coagulation-condensation stages of alkali-activation reactions.
